# Smart Strawberry Farming Using Edge Computing and IoT

**DOI:** 10.3390/s22155866

**Published:** 2022-08-05

**Authors:** Mateus Cruz, Samuel Mafra, Eduardo Teixeira, Felipe Figueiredo

**Affiliations:** Instituto Nacional de Telecomunições (INATEL) Santa Rita Sapucai, Santa Rita do Sapucai 37540-000, MG, Brazil

**Keywords:** Internet of Things, computer vision, machine learning, LoRa

## Abstract

Strawberries are sensitive fruits that are afflicted by various pests and diseases. Therefore, there is an intense use of agrochemicals and pesticides during production. Due to their sensitivity, temperatures or humidity at extreme levels can cause various damages to the plantation and to the quality of the fruit. To mitigate the problem, this study developed an edge technology capable of handling the collection, analysis, prediction, and detection of heterogeneous data in strawberry farming. The proposed IoT platform integrates various monitoring services into one common platform for digital farming. The system connects and manages Internet of Things (IoT) devices to analyze environmental and crop information. In addition, a computer vision model using Yolo v5 architecture searches for seven of the most common strawberry diseases in real time. This model supports efficient disease detection with 92% accuracy. Moreover, the system supports LoRa communication for transmitting data between the nodes at long distances. In addition, the IoT platform integrates machine learning capabilities for capturing outliers in collected data, ensuring reliable information for the user. All these technologies are unified to mitigate the disease problem and the environmental damage on the plantation. The proposed system is verified through implementation and tested on a strawberry farm, where the capabilities were analyzed and assessed.

## 1. Introduction

Strawberries are popular fruits consumed in almost all parts of the globe, and this expansion is a result of the fruit adaption capability that makes the cultivation possible in multiple climates. The efforts of producers and scientists also collaborated to expand strawberry production and commercialization, generating adaptive systems that allow the cultivation of fruit to specific conditions of each region [1]. Furthermore, the production and commercialization of strawberries are present in 76 countries, reaching a global production of 7.7 million tons in 2013. This consumption has increased in the 21st century and has been accompanied by technological innovations that allow the availability of strawberries all year long, thus feeding a continuous market demand [2].

One challenge to be faced is producing the fruit in an economical and sustainable way [1], because conventional cultivation methods apply a considerable quantity of pesticides and agrochemicals to avoid losses by diseases and pests. Nearly half of the world crops are lost due to pest infestation and diseases [3]. For example, Anthracnose (a disease caused by Colletotrichum spp) is the most destructive disease found in Brazilian strawberry farms [2]. As a result, farmers utilize two popular approaches to deal with these problems: prevention and combat. Prevention is associated with good practices such as (i) acquiring seedlings from trusted providers, (ii) controlling water quality, (iii) sanitation and care with the tools of work, and (iv) use of personal equipment [4]. Equally important, combating the disease or plague is crucial when diseases reach a critical state, which occurs when prevention is insufficient or not correctly applied. The majority of combat processes use chemical products to eliminate the disease [4]. However, these products may cause health problems to the consumers and farmers. In addition, these diseases can develop into severe health problems such as auditing loss and congenital malformations through the change in the mother–fetus binomial [5].

Correct disease identification is essential to applying correct products and thus the reduction or elimination of the disease or pest in the plantation, sometimes requiring specialized professionals or previous knowledge from the farmer. In addition, some identification processes must take samples and send them to the laboratory for analysis, consuming more time, leaving the farm unprotected, and allowing disease to spread across other parts [6]. Alternatively, researchers have begun investigating new methods based on Artificial Intelligence and mobile devices, seeking to develop accurate and reliable methods for disease detection. Additionally, techniques that use image-based systems and Artificial Intelligence have begun to be applied because of the considerable visual change caused by most diseases on leaves, fruits, and plant stems.

Computer vision (CV) is a field of artificial intelligence aiming to offer machines vision capabilities. The field of computer vision has presented solutions and applications relevant to the agricultural area, offering autonomous and effective methods of cultivating various different plants [7]. Researchers have widely studied disease control, and it is possible to find in the literature several applications that use computer vision for pest and disease detection [8]. Fruit quality control systems have also been gaining ground in the AI field, causing several applications to be developed for the automatic quality control of harvested fruits [9]. Most of the proposed agricultural applications available in the literature that make use of vision systems often use neural networks to improve processes. Neural network-based systems can detect and classify diseases in real time after a judicious training process. Indeed, diverse diseases can rapidly be recognized, and the farmer can take the correct procedures to mitigate or eliminate the problem early. Several researchers have already developed and proposed solutions that make use of neural networks to detect various diseases in different plants such as corn [10], tomato [11], and many others [12].

This paper proposes a unifying IoT platform with wireless sensor network (WSN), computer vision (CV), machine learning (ML), and long-range (LoRa) communication capabilities. In addition, the paper proposes a cheap and flexible way to implement the proposed IoT platform for heterogeneous scenarios and farming processes. Moreover, the platform makes available to the user all the captured metrics for manual analysis and data-driven decisions. The data collected are stored on Cloud and Raspberry board, consequently creating redundancy in all data and offering different approaches for farmers. In summary, the project aims to be an all-in-one IoT platform to enable the intelligent farm on strawberries cultivation. The organization of the article is as follows. Section 2 presents some background technologies used in platform construction. Section 3 presents a literature review of related work. Section 4 gives a detailed view of the proposed IoT platform. Finally, Section 5 presents the results obtained during the tests performed.

## 2. Background

### 2.1. Wireless Sensor Network

Wireless sensor networks (WSN) are defined as a wireless network of sensors connected to devices such as microcontrollers for monitoring variables or phenomena, allowing users to monitor various and heterogeneous subjects in real time. In addition, the wireless exchange of data between devices also allows a broader range of implementation possibilities since traditional methods that use wires for data exchange may not be applicable in solutions that present long distances between devices. Furthermore, WSNs can address multivariate and unpredictable ecosystems in the agricultural field by monitoring and measuring various physical aspects and phenomena. Correspondingly, the volume of data collected through remote sensing can provide a broad view of agricultural environments and has several advantages through a non-invasive method of collecting information over a large geographical area.

Most sensor network applications seek to develop an autonomous and remote monitoring system of variables such as humidity, temperature, and soil moisture [13,14], and a significant part of the developed systems only read and show those variables on a screen [15]. Some researchers have made progress in this field, reaching more considerable distances using IoT devices [16], automating the farm irrigation system [17], protecting silos from rodents [18], and others. In addition, low-cost wireless sensor networks can be quite vulnerable to external attacks and inconsistent data. Several researchers have developed current solutions to address vulnerability to attack the network and the consistency and quality of the collected data [19]. The proposed platform uses the WSN approach to collect data in the plantation. In addition, the platform uses a machine learning model to detect inconsistent data throughout the collected data and store and display all the collected data. The communication between the devices is done through LoRa technology, ensuring greater coverage and lower consumption in the transmission and reception of data.

### 2.2. Computer Vision

Over the last years, CV applications have gained space in several fields, with deep learning being the most preeminent method. CV applications comprise collecting image resources and identifying them through artificial intelligence techniques. Deep learning allows computational models to learn and represent data, mimicking how the brain perceives and understands information [20]. Deep learning extends classical ML by adding complexity to the model thought layers and transforming the data using various functions. For example, deep learning belongs to the ML field and is similar to artificial neural networks. In contrast, deep learning has the “deeper” aspect of neural networks that supply a hierarchical data characterization by various convolutions. An advantage of deep learning is feature learning, which is the automatic feature extraction from raw data. The disadvantage of the DL approach is the longer training time that it can consume and the computational requirement.

Images are part of the data that can be used to infer the actual status of the farm. Imaging analysis is an important area in the agricultural domain because data analysis techniques are already being used for anomaly detection and classification/detection purposes. Several approaches and computational applications for agriculture have already been thought of and proposed [21,22], including the detection of diseases in plantations. Disease identification tasks use Deep Neural Networks, and several applications have been developed to improve the detection and correctness of the disease on the plantation. In addition, thermal cameras to detect the humidity and prediction of disease on leaves [6], classification of soil conditions [23], and automatic irrigation of the plantation [24] already were proposed.

Robots use computer vision for disease detection and herbicide application. In this approach, small autonomous robots that can monitor and apply herbicides autonomously have already been proposed [25]. The researchers used Artificial Neural Networks for real-time disease detection in the field. The robot uses a Raspberry Pi board for processing at the edge and achieves an accuracy of 96.83%. Another proposed approach uses wireless sensor networks and autonomous robots for disease monitoring. The robots spread across the plantation and autonomously monitor and search for diseases in the plantation [26]. In the proposed solution, the researchers united the entire data collection process with computer vision but in a mobile manner. Therefore, the sensors are not fixed in one place for collection but move around the plantation, collecting and storing temperature, humidity, and light data.

Approaches using Convolutional Neural Networks for leaf disease detection have also been proposed. In [27], the authors presented a new way of using the transfer learning techniques in neural networks to detect plant diseases. This detection happens through characteristics of the diseases present on the leaf. Several architectures were used and results were presented.

### 2.3. Machine Learning

Artificial Intelligence has presented significant advances over the last few years in the most diverse areas. ML is a sub-area of Artificial Intelligence that aims to develop computational techniques related to learning, besides building systems and applications capable of gaining knowledge autonomously. ML algorithms can be used to abstract the understanding of underlying phenomena in the form of a model and predict future values of a phenomenon using the above-generated model. In addition, ML models can detect anomalous behavior exhibited by a phenomenon under observation.

Many papers proposed machine learning techniques on edge using IoT approaches. Moreover, Isolation Forest models have been used to detect cyber attacks on embedded devices. The ML technique analyzes and distinguishes between inliers and outliers values if some man-in-the-middle attack changes the package [28]. Other authors proposed malware detection using Isolation Forest Classification. The proposed system analyzes packets and can detect anomalies in flow information [29]. In fact, most isolation forest applications in IoT aim to capture attacks on devices used in applications and systems.

In the agriculture field, machine learning applications are used for disease prediction. For example, an IoT system that used machine learning capabilities to detect diseases early on in grapes farming is proposed in the literature. The system used the Markov model to analyze and classify the input data based on the crop capture metrics, detecting the disease early [30]. Researchers have also proposed new approaches to Isolation Forest models for outlier detection. Implementing the fuzzy method along with Isolation Forest improves the method’s accuracy and speed in its operation [31].

## 3. Literature Review

Applications of vision systems and Artificial Intelligence techniques for disease detection have already been discussed, and several solutions have been proposed in the literature. For instance, Antonio Oliveira-Jr et al. [32] proposed using a self-supervised technique for leaf disease detection. The system can identify diseases in the leaves of plants such as tomato, citrus, potato, and pear. Several other authors have proposed systems for the diagnosis of specific diseases. Bo-Yuan Liu et al. [33] investigated and proposed using Convolutional Neural Networks for diagnosing the severity of Alternaria Leaf. Alternaria is a group of fungal diseases in plants, which have a variety of hosts and can infect diverse plants. However, the authors focus on diagnosing the severity of the disease present on the apple leaf. Other authors [34] have proposed improvements in disease detection in apple crops using segmentation and disease detection. Segmentation is typically used to locate objects and shapes in images and refers to the process of dividing a digital image into multiple regions or objects. Convolutional Neural Networks have also been applied for classifying diseases present in tomatoes and showed a promising result regarding the performance of the model and the performance of the [35] algorithm.

Researchers have also proposed computer vision applications for Kiwi crop disease recognition that use the YoloX architecture for detection and achieve satisfactory results in their experiment [36]. However, one of the difficulties may be integrating several technologies within a single application. Knowing this, researchers developed platforms to unify heterogeneous technologies while promoting scalability, privacy, and reliability in data and processes [37]. The possibility of integration between applications brings the possibility of using data from sensors and images together, making better use of the variables offered. Wireless sensor networks have already been proposed in the literature to make planting and cultivation sustainable and, at the same time, more intelligent so that the data collected in the field are used for improving planting and production [38]. Some wireless sensor networks have a particular monitoring variable such as the one proposed by Jaime Lloret et al. [39] with soil mixture monitoring as its central focus. In fact, the soil and its characteristics have a vital role in the quality of the fruit and the health of plants, and periodic maintenance should be done regarding nutrients and minerals.

Applications of computer vision for disease detection, unified platforms, and wireless sensor networks have already been extensively proposed in the literature, but merging all of these into one more strawberry-specific application has not been found. Therefore, an IoT platform using Yolo v5 architecture, WSN, and LoRa technology was developed to detect strawberry diseases and remotely monitor ambient conditions using long-distance and low-power communication technology.

## 4. Proposed IoT Plataform

This work developed an IoT platform that used Raspberry Pi 4B, Arduino, Sensors (DHT11, hygrometer, and camera), and a LoRa module. Briefly, the platform that can be found in the Github repository (Github of the proposed platform: https://github.com/matteuscruz/Smart-Strawberry-Farming-using-Edge-Computing-and-IoT, accessed on 15 July 2022) uses sensors connected to Arduino for collecting local data and sending it to the Collector node. All data exchanged locally between the devices uses LoRa communication technology, while other services such as uploading and storing data in the cloud make use of the 802.11n standard. Figure 1 illustrates the entire solution and their perspectives layers. The entire platform has four layers, each corresponding to specific tasks. The L1 is composed of sensor nodes, and L2 is composed of collector nodes. Web services are deployed in L3, and user applications are deployed in L4.

Layer one is responsible for collecting the local data, pre-processing, and transferring. Processes in this layer are entirely local and cover all local data collection and data transfer. The Arduino board is responsible for correctly reading and structuring the data collected by the sensors, using the JavaScript Object Notation (JSON) format to send the data. Layer two is responsible for all data processing, organization, storage, and uploading. The training of the CV model is done via Google Colab. The Yolo v5 and Isolation Forest models is stored and run locally on the Raspberry Pi board. As for the ML model, training is also done off-board, using only the Scikit Learn library and the already trained model to run the application. The data are first interpreted by the LoRa receiver, processed and stored locally by the Raspberry Pi board, and sent to the cloud via the Wi-Fi connection.

Layer three is an intermediary between local and remote tasks running on the platform. The alarm task is done locally in real time, but once activated, it requires cloud services to be completed. In an analog way, online and offline dashboards were developed, thus allowing more coverage of possible application scenarios. Finally, layer four offers users access to applications and services offered by the platform. Therefore, the layer is responsible for providing the user with all access to the services provided within the IoT platform. This layer provides Virtual Network Computing (VNC) access to the CV application, reading and analysis of the collected data, and configuration of the collector node operation mode, among other resources.

### 4.1. The IoT Platform

The platform uses two wireless technologies, one for device communication and another to upload the data to the internet. Data exchange between the devices is performed by LoRa, while the data upload to the internet is over Wi-Fi through the Message Queuing Telemetry Transport (MQTT) protocol. LoRa can reach large distances with low energy consumption but has a narrow channel for transferring data. Wi-Fi in the collector node is necessary to upload data to the internet, providing a cloud dashboard and storage system that can be accessed everywhere by the internet. Figure 2 illustrates each layer of the entire architecture.

The CV application runs at the collector node Raspberry Pi board, where a camera is connected, and an internet connection is available. The camera is necessary for capturing real-time images from strawberry plants, and an internet connection is used to guarantee access to the real-time detection completed by the detection model. A more detailed vision of the pipeline utilized on CV application is illustrated in Figure 3.

The camera is installed at the perception layer for capturing and sending real-time images to the Raspberry Pi board. The detection layer contains a Yolo v5 model, an internet connection, and a VNC server installed. The visualization layer offers remote access to Yolo v5 output.

#### 4.1.1. Detailed Overview of the Hardware

One aim during the development of the sensor nodes is to choose the hardware components that make it easy to replicate the project. However, another important aim of the project is to be suitable for heterogeneous farms. This way, the platform has been built and elaborated with prototype boards (with easy development and use purposes) and plug-and-play approach sensors.

The Raspberry Pi is the main component of the platform, which is responsible for edge CV and machine learning tasks. The CV system is responsible for executing the Yolo v5 locally and consequently making detections and classifications of the diseases. Moreover, an ML model searches for outliers and errors in sensors’ local readings using an Isolation Forest model. In addition, the board offers a Wi-Fi connection natively, acting as a gateway for uploading the collected data from the sensor nodes sent by LoRa. A Universal Serial Port (USB) High Definition (HD) camera is also present on the board, serving as an input for the detection model. Capturing images for the neural network can be made by a Camera Serial Interface (CSI) Raspberry Pi compatible camera or a USB webcam. A USB HD camera is used in tests made on the field. Some components used on the collector node are illustrated in Figure 4.

The main components of the collector node hardware are:Raspberry Pi 4B: A only computer board with 4 GB of RAM, Broadcom BCM2711, Quad-core Cortex-A72 (ARM v8) 64-bit SoC @ 1.5 GHz. The board presents a native Wi-Fi connection.Radio Frequency Wireless module LoRa 915 MHz [flip-flop]. A Lord module that operates in the 850.125–930.125 MHz range, with reception sensibility of −147 dBm and transfer range of 2.4–62.5 Kbps.HD USB Camera: An 1080p camera compatible with Windows and Linux systems, USB 2.0.

The sensor node comprises an Arduino board, LoRa module, batteries, humidity, and temperature sensors. Arduino handles the systematic data collection from the sensors and sends the data to the collector node via the LoRa link. The Arduino is chosen for four reasons: (i) price, (ii) availability, (iii) community, and (iv) learn curve. The board has a microcontroller that does low-level control of the GPIO (General Purpose Input Output) included, facilitating the implementation of the project. In addition, the development and deployment of the platform through Arduino facilitate the replication and implementation of the platform. The board also had a strong community of developers across the web that uses and supports diverse projects in fields such as automation, robotics, and the Internet of Things. The construction of the sensor node follows an easy-to-implement approach, presenting an easy and quick approach. The sensors also use the power supply from the Arduino board pins, not requiring another power source. Figure 5 illustrates the components of the sensor node, which is used for capturing local farm data. A detailed description of each component is given below:Humidity and temperature sensor DHT11: It is a sensor of relative humidity and environment temperature. The sensor can capture relative humidity in a range of 20 to 90%. DHT11 also captures temperature on 0 to 50 °C with a precision of ±5.0% UR and ± 2.0 ºC with a time to respond equal to 2 s;Soil humidity sensor hygrometer: Soil humidity sensor is based on LM393 comparator with easy installation, low price, and large availability;

All the components have great availability and a low price, making it easier to implement and develop the platform. A summary of the configuration of collector and sensor nodes as well as the connection are illustrated in Figure 6.

#### 4.1.2. Detailed Overview of Software

The aim is to implement an IoT platform with the respective resources: (i) local data collection, storage, and visualization, (ii) reliable data approach, and (iii) fog disease detection. The project must deliver the farm conditions in a friendly way, making it possible for non-specialized users to comprehend the farm scenario. Because of that, a friendly and customizing dashboard is used to show users the captured information.

The software behind data-collecting tasks is initialized in sensor node with Arduino sensors readings, mediated by Raspberry Pi processing and upload, and finished in Cloud Application Programming Interface (API). The Arduino scratch reads, interprets, and structures all the collected data in JSON format. After, the data are sent to the LoRa module through the three and four Arduino General Purposes Input Outputs (GPIO) in string format. The LoRa module sends the message to a predetermined address. On the other side of the link, the collector node waits for some transmission. In order to organize all the data, a specific period is reserved for each node. In this way, each node needs to wait for her time to transmit. For energy-saving purposes, no activity is done by the microcontroller during idle time. Figure 7 illustrates how the multiplexing schema runs to allow multiple nodes to work in the same channel.

Data reaching the collector node are first sent to NodeRED flux through the MQTT protocol. NodeRED is the main WSN software component in the collector node. NodeRED is responsible for machine learning functions, data structuring, and service connection. In order to structure the data for machine learning digestion, a JSON file is created with temperature and humidity data on the NodeRED platform. After the JSON data are injected into ML input and classified, the output is sent offline and to the Cloud database.

InfluxDB is the database service used for storing all the captured data and classifications completed by the IoT platform [40]. The database gives the user the option of storing the data in the Cloud or locally. The primary motivation for using the platform is in the time series format present in almost every task due to their natural temporal ordering, including temperature and humidity. Other reasons are its performance on the Raspberry Pi, offline availability and Cloud options for storing data, and additional integration with NodeRED and Grafana. The time-series approach is essential to capture environment variables from diverse plantation areas, where each data snapshot represents a unique farming condition. The sensor node collects the information from the sensor, sends it to the collector node, retrieves a timestamp, and stores the timestamped data on a local and Cloud database.

The InfluxDB database is connected to Grafana through an API, presenting the collected data in a customized, clean, and concise dashboard. Grafana is an open-source analytic and monitoring solution for the database [41], and it is chosen because of the stronger customization possibilities for the dashboard, clear dashboard visuals, and integration with InfluxDB. In addition, the service offers numerous tools to deeply analyze the data such as tendency, median, mean, and standard deviation. Furthermore, the numerous types of graphics offered by the platform to plot the captured data are essential to extracting different collected metrics, such as temperature and humidity tendency, over the month. In addition, all the machine learning outliers detections are plotted in a bar format, representing how many wrong sensor readings were captured through time.

Access to the dashboard can be made from a smartphone, tablet, and laptop with internet access. All devices offer access to the same dashboard plantation metrics in different formats. The dashboard follows a web-based approach, not requiring the installation of any mobile application to access all the information collected on the farm. On the other hand, a user and password credential is necessary to access and modify the dashboard and database. In order to better present the proposed dashboard for mobile devices, Figure 8 illustrates the dashboard view from smartphone and laptop.

### 4.2. Computer Vision System for Disease Detection

This section describes the CV system deployed on the collector node. In addition, we explain how the model is generated, training and test phases, and parameter configurations. The CV system can be divided into three central layers: (i) perception and image capturing, (ii) model ingestion and detection, and (iii) user access to CV detection. The first two layers are processed locally in the collector node, and the third can be processed remotely or locally by the user. Furthermore, the development of the CV application for the platform has three main objectives: (i) a way to implement and use a high-accuracy CV model in remote locations with no internet connection, (ii) real-time strawberry diseases detection, and (iii) identify seven of the most common diseases in strawberry farming. The final deployed model can detect and classify seven of the most common diseases in strawberries. Figure 9 [42] illustrates in more detail the presence of these diseases in strawberries. These diseases are described as follows:**Angular Leaf Spot**: Known as “bacterial stain”, the disease is caused by *Xanthomonas fragariae* bacteria. The name is given by the appearance of light green, increasing their size later until they become visible on the sheet.**Anthracnose Fruit Rot**: It is a rot caused by agents such as *Colletotrichum fragariae*, *C. acutatum*, and *C. gloeoporioides*. The rot is generated when the plant is submitted to high temperature and humidity conditions.**Blossom Blight**: Disease symptoms began with gray sporulation of the fungus on the stigmas and anthers of strawberry flowers, followed by necrosis and complete flower abortion.**Grey Mold**: Caused by the *Botrytis cinerea Pers. F*. fungi or simply *Botrytis*. The name Gray Mold is given to the appearance of gray color mold formed in the leaf and calyx, which may affect the fruit.**Leaf Spot**: Known as “*Mycospherella spot*”, the disease is caused by *Mycosphaerella fragariae* fungi, and it is one of the most common diseases on the strawberry. The fungi attack mainly the leaflets, presenting a purple color that posterior develops into a brown color.**Powdery Mildew Leaf**: Caused by the *Sphaerotheca macularis* fungi. The powdery mildew is caused when the plantation is submitted to a dry and hot climate. Manifestation is given by the apparition of whitish spots on the inferior face of the leaf. It also affects the fruits and can be observed by the fruit’s discoloration and spots.**Powdery Mildew Fruit**: It is caused by *Sphaerotheca macularis* fungi, affects the fruits, and can be observed by the fruit’s discoloration and spots.

In order to reach all these objectives, some requirements must be met. First, a lightweight CV infrastructure must be used, ensuring good performance in embedded systems with low computer power hardware. To meet this requirement, Yolo v5 architecture is deployed on the board. Yolo v5 is part of object detection architectures, pre-trained on the COCO dataset, and represents Ultralytics open-source research into future vision AI methods [43]. A good dataset, parameter configuration, and training are necessary to guarantee a high-accuracy model. The dataset chosen is available on the Kaggle community website and is named the “Strawberry Disease Detection Dataset”. The dataset consists of 2500 images with the corresponding segmentation annotation files for different types of strawberry diseases.

#### 4.2.1. Yolo v5 Architecture

Released on 18 May 2020, Yolo (acronym to You only look once) is one of the most famous object detection algorithms developed. The algorithm uses an one-stage object detection method [43]. In addition, Yolo v5 is the last architecture between various releases by Ultralytics. Equally important, this version presents a better accuracy in detection and a lower time to make inferences in predictions. Then, the main aim of the network is the possibility of running on less powerful devices such as Raspberry Pi boards and smartphones. In addition, there are four versions of the Yolo v5 network: (i) Yolo v5s, (ii) Yolo v5m, (iii) Yolo v5l, and (iv) Yolo v5x. Each version has its own characteristics, Yolo v5s owns among the other versions the best speed to calculus but has the lowest average precision among them. In brief, the version chosen for deployment in our system is the Yolo v5s, because this version presents models that are 75% smaller than the others, faster recognition times, and better accuracy.

The Yolo v5 architecture can be divided into three main parts: (i) backbone, (ii) neck, and (iii) output. The backbone is composed of a Convolutional Neural Network, which is responsible for capturing the characteristics of the image. The images captured by the camera go first into the focus structure, which is on the network backbone. The backbone first layer has the function of reducing calculations made by the model, speeding up the model training process. The image with resolutions equal to 640 × 640 × 3 will pass through a slicing operation that will reduce the resolution down to 320 × 320 × 12. Finally, via a normalization batch layer and the Hardwish function, the results are presented on the last layer.

#### 4.2.2. Dataset and Training Phases

For training purposes, a dataset with 2500 images is used to train and test the detection model proposed. The dataset founded on the Kaggle community website is created, labeled, and offered by members of the AI lab, Computer Science and Engineering department, JBNU [42] and has images of strawberries with seven major and most common diseases: Angular Leaf Spot, Anthracnose Fruit Rot, Blossom Blight, Grey Mold, Leaf Spot, Powdery Mildew Fruit, and Powdery Mildew Leaf. The 2500 images are split into 80% for training and 20% for the testing phase.

A significant advantage of using labeled datasets is the rapid training and implementation of the model in the field. Consequently, capturing and manually labeling all the images comes at no cost. These tasks can also be more time consuming on all the pipelines to develop an object detection model. Websites such as Kaggle provide a large collection of datasets of images and raw data that can leverage artificial intelligence applications. In addition, several professionals and universities contribute to the community, making several datasets available to the public. Intending to demonstrate the developers labeling method for dataset development, Figure 10 shows the ground truth training data for training purposes.

Raspberry Pi 4B boards present limited hardware capacities for training purposes. Therefore, a computer with a powerful Graphic Process Unit (GPU) is required to expedite the training phase. Training time and success are intimately correlated to GPU and hardware used during the training phase [44]. Because of that, a cloud tool developed by Google named ’Google Colab’ is used. The service offers access to high computational power servers with GPU to write and run Python codes through notebooks for free.

To train an accurate object detection model, hyperparameters must be configured and training well executed. Some parameters must be set to configure the training process duration and execution. Epochs and batch size are parameters that configure the extension of training. Batch size defines the number of samples to work before the model parameters tuning. Epochs are related to controlling the number of complete passes of the model through the training dataset. In addition, Yolo v5 offers diverse weight sizes that affect the performance and accuracy of the model. Epochs are fixed at 100 and the batch size is fixed at 16 for the proposed Yolo v5 model. This is because Google Colab limits the use of the server, and this configuration requires less than three hours to complete. Moreover, the weight size of the Yolo v5 model is iteratively changed during training. Table 1 [43] presents the weights size available for model training. Briefly, Yolo v5n and Yolo v5s have speed but low accuracy. In contrast, Yolo v5m, v5l, and v5x have less speed but more accuracy than others.

### 4.3. Machine Learning

An ML application is developed to detect errors in sensor readings. The application uses the Isolation Forest Classifier algorithm, which is a tree-based anomaly detection model. The algorithm considers anomalies as data that differ from typical characteristics [45]. Standard characteristics are those normally exhaled within a process, and outliers are unexpected data. Specific regions exhibit well-known temperature profiles, so outliers would be any outliers presented by the sensor node. These values are usually the result of sensor, reading, or connection failure.

The reasons behind the choice of the algorithm are the small memory requirement and low linear time complexity [45]. All this assures that the model will be performative in real-time data analysis on a Raspberry Pi 4B board. The model installed in NodeRED analyzes real-time data. That function provides more reliable data for user information about conditions to which strawberries are subjected. In addition, this approach considers that sensors readings can fail at any time, putting at risk all the IoT platform data-driven approaches.

#### Implementation and Working Pipeline

Isolation Forests is an anomaly detection model that uses a dataset where the model target contains few samples among so many standard data. The few samples are denominated outliers. The goal of the model is to build trees to isolate these anomalies. Briefly, the Isolation Forest is a set of Isolation Trees, similar to that seen in Random Forest [45].

Isolation Forests is an anomaly detection model that makes use of a dataset where the target of the model contains few samples among so much normal data. The goal of the model is to build trees to isolate these anomalies. The model is part of the Scikit Learn library [46] and deployed into the NodeRED dashboard tool. The tool allows the development of an ML flow for real-time ingesting sensor data and verifying outlier presence. The development and training phases are done locally using Anaconda Navigator and Python. First, a dataset is defined for training purposes. The dataset consists of temperature and humidity readings captured in the southwest region of Brazil. The use of correct data for training is essential to ensure that the model captures the correct data characteristics. Any data that do not have standard characteristics can be classified as an outlier. The model parameters are listed below:**Bootstrap**: False;**Contamination**: Auto;**Maximum Features**: 1.0;**Maximum Samples**: 100;**Number of Estimators**: 100;**Number of Jobs**: None;**Verbose**: 0.

The input of the ML model is provided by sensor nodes responsible for collecting and structuring the captured data. All the data structured in JSON format are injected into the model input in real time, making the system automatically analyze and search for temperature and humidity changes. The model output is sent to the InfluxDB cloud and offline database. The Grafana API obtains the data in real time and plots the information in the dashboard into a bar graph for the user. All outliers detected can be consulted in a database or dashboard. Additionally, problems related to sensors can be more easily detected with applications with dozens of sensor nodes.

### 4.4. Encapsulation

Agriculture applications demand a hard case to protect the hardware components from climate change. At the same time, periodic maintenance needs to be the minimum possible for two reasons. First, the devices can spread a considerable distance between them. Second, many devices can make it difficult or impossible to repair regularly. Because the platform is being developed for operation in outdoor plantations and is subject to extreme environments and situations, encapsulation is more necessary so that every electronic part of the project is fully protected from (i) dust, (ii) moisture, (iii) friction, and (iv) rain. Therefore, one case is developed to protect the platform and another is developed to protect the sensor nodes.

Some features were taken into account. First, the protection case needs to accommodate all the electronics and batteries. Second, the protection case needs to accommodate the antenna of the LoRa module and the sensors on the outside, because electronics heat can influence the environment’s measured metrics. In addition, the radio signal of LoRa can be mitigated by protection. Another essential feature of protection case development is the material that is constructed. Furthermore, another important aspect is the construction material. Large deploying can add some environmental impacts because of the addition of plastic, metal, and other materials used on the project. Consequently, the case is made of polylactic acid (PLA) to mitigate the environmental impacts of large-scale deployments.

PLA is the second most-produced biodegradable plastic. PLA is an excellent option for the chemical versatility that presents good form stability. The material has good stability through temperature change, presenting deformation only above 80 Celsius degree [47], making it suitable for the application. The following phase is to bring the design to the physical world. For this, 3D printing is used. In brief, 3D printing is the process of joining materials to make objects layer upon layer from 3D model data. At its beginning in the 1980s, the first applications were prototyping, but in the following decades, the manufacturing technologies and materials transformed, and new fields of application appeared [48]. The wrapper proposed is developed through Fusion 360 software, where all the necessary adjustments are made.

A design model needs to be created to print a case for the devices physically, and Autodesk Fusion 360 is the software used to create the design file for the protecting case. The software offers quick and easy modeling tools, enabling a fast prototyping schema. Two design models are created on the software, one for the collector and another for the sensor node. A custom case allows the sensors, LoRa antenna, and camera devices to remain in a convenient location while the rest of the circuitry remains protected. The 3D printing used is the Creality Ender Pro, and the slicer utilized is the Ultimaker Cura, the software supports many formats, but the STL format is the most common, widespread, and used by the maker community. Consequently, this is the format utilized for the proposed case. Figure 11 illustrates which final design was chosen for the development of the cases.

After developing the 3D model file, the 3D printer must be configured to create a solid and hard case for deployment. Cura software offers a lot of configuration tools for printing objects. At the same time, it offers a basic configuration system for non-expert users. The Creality Ender 3 Pro takes 13:24 h to print the body and more than 5:46 h to print the cover for each sensor node. In addition, the collector node takes 13:45 h to print the body and the same time for the cover because the cover design is the same for both. In addition, two sensor nodes and one collection node case protection are created, resulting in 57:51 h to print all. The final result can be seen in Figure 12.

### 4.5. Cost Analysis

The proposed IoT platform uses several hardware and software components to provide all the necessary functionality. The hardware components present the highest cost for developing the sensor and collector nodes, since most of the software is or presents free options for their usage. Thus, the total cost to develop the platform was calculated, being all products quoted from reliable suppliers and recommended by the brands. Table 2 presents the items, their values, and the quantities.

The construction of the collector node is the most expensive process in platform development. However, the development of only one collector node may be sufficient to support multiple sensor nodes deployed over the farm. The lower cost of sensor nodes enables greater application scalability. This way, the farmer can affordably and efficiently monitor several processes and locations simultaneously. Other costs in developing the platform are related to the software used. However, a large part of the software offers no additional cost to the farmer or offers free option plans. Preferring not to incur these costs, the farmer still has the option of not using any of the tools offered. Most of the tools presented operate independently and do not require the use of the tools together to execute their functionalities. The only set of software that must be implemented together is the InfluxDB database and the Grafana dashboard if the user chooses to present the information in Grafana. This is due to Grafana API capturing the data directly from the InfluxDB database for display. Table 3 summarizes all the software used.

Using LoRa technology for communication between the devices lowers the final cost of the platform since no bandwidth licensing is required. Several other tools can be used to reduce the cost, but none of them were considered in the proposed development. In summary, all logistical costs, app development, installation, and maintenance are disregarded in the first moment. The platform offers a considerable cost and can be used by small and large producers.

## 5. Results

### 5.1. Case of Study

This subsection is dedicated exclusively to the field testing of the IoT platform, presenting materials and methods used for implementation and testing. The validation was performed on a strawberry farm located in a rural area. The entire plantation is approximately 18,896 m^2^ in size, and an area view of the entire plantation area can be seen in Figure 13. The plantation has a total of 104.000 San Andreas species planted. The cultivation began on 14 March 2022 because San Andreas strawberry is a moderate day-neutral with a production pattern. Two sensor nodes and a collector node are deployed to cover the area. The sensors nodes collect the local temperature and humidity condition and send thought LoRa communication to the collector node. The collector node uploads all the data to the internet, presenting the information on a dashboard, analyzing the climate condition of the plant, and storing all the data on an online database.

First, the location of the sensor nodes needs to be chosen. The positioning schema of the sensor nodes also needs to be sufficient to cover the most critical parts of the plantation. The strawberry farm chosen has a small area, and because of that, two sensor nodes are sufficient to validate the IoT platform. Second, the variables collected need to be relevant to the plantation health and quality. As well as many other plants, strawberries are fragile fruits that take damage by some heat and humidity conditions. Because of that, the DHT11 and hygrometer sensor are chosen to validate the Machine Learning algorithms.

To finish, the real-time capture and detection performed by the CV application need to be validated. The CV application can detect seven disease types on the strawberry. However, finding these diseases on a unique farm is challenging. In order to find strawberry specimens that contain the diseases present in the dataset, a small plantation area was selected and continuously presented in real-time to the collector node.

The test roadmap for the IoT platform is written as follows: (I) implementation of the sensor nodes, (II) implementation of the collector node, (III) verification of the connection between the nodes, (IV) collection and access to the collected data, (V) storage and visualization of the data, and (VI) validation of the computer vision model in the field. Sensor nodes are deployed on the two middle points related to the center and borders of the plantation. This approach maximizes the coverage area through three points and good platform deployment. Nevertheless, the sensor nodes can be dispatched to the more critical areas of the plantation, covering the most exciting points.

LoRa communication is used because the local does not offer Wi-Fi connectivity. The online dashboard and database also cannot be used by a lack of internet connection. In addition, offline dashboards and databases are used as an alternative. Because of that, all the data are collected but available only locally. No wireless LAN was created for the application tests, so a notebook is used to check the collected data and the detections performed. The goal of the real-time test is to validate the model ability to perform processing entirely at the edge, enabling its implementation in smart greenhouses or autonomous rovers. Figure 14 shows the connection schema applier for test purposes and Figure 15 illustrates a possible application of rover.

Since the application will be executed at the edge through the collector node, a rover can be used for autonomous monitoring through the plantation. In addition, the collector node can also be used in vertical plantations to monitor and detect disease. Otherwise, the platform is useful when the farm does not count on the internet. This means that all the data are centralized on the collector node and available locally only. This is necessary to serve a larger number of users since several farms are located distant from cities and connections. In addition, this makes it possible for the platform to be implemented in more scenarios. Figure 16 shows the sensor node implanted on the plantation.

The sensor nodes were the first to be deployed along the plantation. The deployment locations were strategically selected considering the maximum distance between the collector node and the sensor nodes. The local dashboard displayed on the computer is responsible for clearly presenting all the data collected by the sensor nodes implemented on the plantation. The presented data are also saved in a local database to guarantee future queries by the user. All data sent by the sensor nodes are centralized in the collector node in order to maintain a greater organization of the captured data for presentation and storage. To organize the data, a Python script always keeps the collector node waiting for new data sent by the sensor nodes by LoRa connection. After, the data are forwarded to local NodeRED. The offline dashboard is created on NodeRED and displays all the metrics captured (see Figure 17). The sensor nodes names can be personalized on the platform, but test numbers are designed for the name.

The CV application is tested with a quick walk with the collector node. An implementation in a real scenario analyzes the deep learning algorithm detecting capabilities in real time. The test consists of a search for some signal of disease in strawberries and pointing the camera at the leaf. The learning model can detect the diseases present on the plantation. In the field test, the detected diseases are Angular Leaf Spot and Leaf Spot. Figure 18 shows the outputs presented by the model:

Detection is completed in real time through a deep learning model installed on Raspberry Pi. A notebook is connected through an Ethernet cable to the Pi board to see the detections. In real platform implementations, the pi screen needs to be streamed to a smartphone or laptop screen through a VNC connection over the internet.

#### Computer Vision

The performance of the proposed model is evaluated through metrics such as (i) mean average precision, (ii) precision, (iii) recall, (iv) accuracy, and (v) box loss. A detailed description and the equation of all these metrics are shown above:**Mean average precision (mAP)**: The metric presents the mean of the average precision (AP), in which AP is represented by the area below the curve.**Precision**: Relation of all the classifications by the model to the correct ones. In general, it is used in situations where false positives have more weight than false negatives;
(1)$$Precision=\frac{TP}{TP+FP}$$**Recall**: Relation between all the true positives by the sum of true positive and false negative. It is like precision but used in situations where the false negatives are considered more harmful than the false positives.
(2)$$Recall=\frac{TP}{TP+FN}$$**Accuracy:** Tells about the general performance of the model, being the relation of the correct by all the classifications performed by the model. *Accuracy* is a good general sign of the performance of the model.
(3)$$Accuracy=\frac{TP+TN}{TP+FP+TN+FN}$$**Box loss**: Measures how close the bounding box is from the true box.

First, metrics regarding the model performance during the training are captured through a training dataset. A training dataset is a set of data used during model learning, making parameter adjustments such as weights to ensure better detection and classification capabilities. The metrics presented above are used during the training phase to follow the performance of the model during each epoch. Each epoch represents one time that the model ran through all the images. Moreover, all the metrics are applied in different weight sizes to search for a better and more performative model. In addition, all the tests between the different weight sizes are summed up in Table 4.

In brief, all the models performed in a similar way under the same hyper-parameters, and some detections made on the training dataset can be seen in Figure 19.

However, models with large weights consume more space on the SD card and have little performance difference to the final result. Because of that, the Yolo v5s is the chosen model for the task of disease detection on a strawberry farm. The model also showed promising results on the test dataset. A test dataset is a set of data independent of the training dataset and is an excellent way to test the model detection capability in different scenarios and situations. The performance presented by the model on the test set shows satisfactory results with a mean Average Precision (0.5) of 78.7%, Accuracy of 92.8%, Recall of 90.0%, and an F1-Score of 76% as shown in Figure 20. The confusion matrix presented in Figure 21 also presents satisfactory results of the model, presenting a good ability to detect the model in six diseases. However, the model presents some problems recognizing the Angular Leafspot disease, scoring only 0.23 in the obtained matrix.

Several analyses have been performed in order to find the reason behind the performance of the Angular Leafspot class concerning the other classes. The first test consists of applying the Gradient-weighted Class Activation Mapping (GRAD-Cam) technique to the model to see which features were considered for the final classification. In summary, GRAD-Cam is a technique that produces visual explanations of decisions in CNN-weighted models. Through this technique, it is possible to observe through a heat map which regions in the image are the most important for the model classification and prediction processes. However, during testing, the model could not accurately identify the disease’s main features (the spots created on the leaves) in all detections. Figure 22 shows an example of which features were considered relevant in classifying the Angular Leafspot disease.

In further tests using GRAD-Cam on the other classes, it is possible to see the model’s accuracy in determining which regions present relevant characteristics of each disease. For example, in the Powdery Mildew Fruit disease, the model could perceive which visual changes presented in the fruit were the results of the disease. Figure 23 presents an example of GRAD-Cam applied to the Powdery Mildew Fruit class.

The dataset was also analyzed, as unbalanced examples between classes in a dataset can cause different performance between classes. The first step quantified how many samples (images) of each class were provided to the model during the training stage. A summary of the information found can be seen in Table 5. Although having an unbalanced dataset, Angular Leafspot has a considerable number of images when compared with the other classes. Thus, tests with a larger dataset and new images should be performed to prove their impact on the model results.

Since this is an application developed to be run in the field, the brightness variation in the captured images can change the detections performed by the application. With this in mind, a test was made to present several images to the model with different brightness profiles to test the model performance in several brightness variations. For that, a Python algorithm that iteratively changes all image brightness in the test dataset was used. An illustration of the final images presented by the model can be seen in Figure 24.

The model achieved considerable performance even when darkness was 40%, while difficulties appear when the model is exposed to images or scenes with low lighting, where the disease detection is impaired. Table 6 shows what precision, accuracy, and recall were obtained from the model under different brightness conditions, and Figure 25 shows a sample of the detections performed by the model in the different dark conditions.

A new dataset with images with different brightness profiles has been developed to solve the model errors in different lighting conditions. This new dataset was used to retrain the model using the parameters described in the previous steps but now with four times more images. In the end, the model was presented with images with different lighting configurations and could detect the diseases in them accurately. Figure 26 demonstrates the ability of the model to detect and classify diseases in different lighting profiles. An LED next to the collector node pointing directly at the plant may also be also enough to avoid problems with illumination.

Finally, another field test was performed at different times during the day to evaluate the model ability to detect and classify different diseases in different illuminations. Four schedules were defined for capturing and presenting images for the proposed model. The chosen times should be aligned with the test proposal and the conditions offered by local farmers. So, the test times were defined as 7:00 a.m., 9:00 a.m., 2:00 p.m., and 5:00 p.m. At 7:00 a.m., the light conditions are more sparse, offering a higher challenge for the model to discern the various diseases that can be detected. At 9:00 a.m., local light conditions are more abundant, and the model appears to have higher confidence in its detections. At the same time, at 2:00 p.m., the environment is so bright that model performance is impaired. Finally, at 5:00 p.m., the color change due to sunset significantly impacts model performance, causing some specimens with diseases to go unnoticed by the detector. Some detections made by the model can be seen in Figure 27.

Benchmarking of the proposed model performance on edge has also been completed and the metrics captured. The model runs on the edge through a Raspberry Pi 4B board with a Broadcom 2711 Quad-core Cortex-A72 64-bit SoC @ 1.5 GHz and 4 GB of RAM. Both processor and RAM were monitored at 1-minute intervals for a total of 1 h, resulting in 60 samples of each. The temperature displayed by the board, frame rate and processing time displayed by the model, and total processor usage by the board were also observed. However, Equation (Equation 4) was considered for calculating the frame rate presented by the model. Figure 28 displays the behavior of the board running the detection model in real time.
(4)$$Frame\;Rate=\frac{1}{Processing\;Time}$$

Some improvements must be made in the collected node, the first being the addition of some active heat dissipation method due to the high temperatures reached during the execution of the computer vision application. The temperature achieved of 85 °C is critical for the Raspberry Pi health if it is maintained for long periods. Another point to be considered is the significant processing time presented by the model, which on average is 2 s. Thus, the model improvements are necessary to reduce its need for computational power and thus increase its performance.

Briefly, the model achieved an accuracy of 92% on the training set and 92.8% on the training dataset. The model also maintained good quality in its detections when used in real time in crop applications and made accurate detections even in different illuminations.

### 5.2. Machine Learning

The dataset presented to the model is split into 80% for training and 20% for testing. The outliers were generated through a uniform distribution function in Python language. The function generates values between 0 and 50 for temperature and values between 0 and 100 for humidity variable. After, these values are appended to the original dataset. The model obtained an accuracy of 93.562% on the test set in the tests performed. Plots illustrating the temperature, humidity observations, and the outliers generated are shown in Figure 29 and Figure 30.

In order to better understand the decision criteria used by the model to detect outliers, the Shapley Additive exPlanations (SHAP) framework presented by Lundberg is used, which is based on Shapley values and built on concepts of game theory. The framework aims to use Shapley values to describe how much each predictor contributes to the model final result. The framework makes use of Equation (Equation 5) to obtain Shapley values [49].
(5)$$\phi_{i}=\sum_{S\subseteq F\setminus \left\{i\right\}}^{}\frac{|S|!(|F|-|S|-1)!}{|F|!}\left[f_{S\cup \left\{i\right\}}\left(x_{SU\left\{i\right\}}\right)-f_{S}\left(x_{s}\right)\right]$$

The model has only two alternatives as output, inlier or outlier. These values are calculated from the Shapley values obtained where the marginal contribution of each observation is i. If a given observation is detected as an outlier, the model output will be ‘−1’, while observations detected as inliers will be ‘1’.

The SHAP framework offers several tools for generating interpretable graphs. These graphs show what impact a given variable had on the final model ranking based on the model input observation. This impact, in turn, considers the Shapley value for relevance within the ranking. Figure 31 illustrates eight classifications made on samples from the dataset.

From this sample, it is possible to observe the importance that each value and variable had within the final classification made by the model. For example, given the training sample presented to the model during training, temperature values much higher than expected can be classified as outliers. Similarly, given the temperature sample presented to the model during the training stage, values much lower than expected may be detected and classified as outliers. The SHAP framework offers several tools for generating interpretable graphs. These graphs show what impact a given variable had on the final model ranking based on the model input observation. This impact, in turn, considers the Shapley value for relevance within the ranking. In summary, the implemented model can detect outliers presented and distinguish between the actual temperature and humidity values and not.

## 6. Conclusions

This article proposes an intelligent Internet of Things (IoT) platform. In addition, the platform was validated through practical application on a strawberry plantation. The platform offers several opportunities for the implementation of a digital farm. In addition, CV, ML, and WSN technologies are included on the IoT platform. The proposed platform has a focus and is validated on strawberry farming. In contrast, flexible architecture and construction offer other opportunities for modifications and implementations in other plants. The CV model can be retrained to detect other diseases related to other plants. The ML model can also be retrained to fund outliers in different temperature and humidity scenarios. All WSN architecture can be reused for other farming processes because the data collection process is not linked to the fruit produced.

The sensor nodes collect essential metrics that influence plant health, such as temperature. The CV model is tested and detects diseases on edge, not requiring an internet connection. However, at sunset, the model did not detect all the diseases presented and maintained a low frame rate, requiring improved performance in different lighting profiles and better performance at the edge. All these suggestions remain for future work. In addition, 3D-printed cases offer hardware protection and durability to the platform. On validations, the case protected the hardware from dust and humidity on the farm for a brief period. However, sensors such as DHT 11 profs are not prepared to deal with the agricultural environment. Because of that, other sensors need to be considered in real applications. The 3D printing approach in case construction allows users to adjust for other sensors.

LoRa communication is also validated in the practical scenario. The LoRa multiplexing schema is validated, and all packages seeded at the sensor node were successfully delivered. Only one channel is used to connect multiple sensors, lowering the cost of implementation and not requiring a high-cost LoRa gateway to upload the data to the internet. Package collision is avoided through more significant time intervals between device publishing. Slow changing profiles of environment variables allow working with multiple sensors with large intervals between them. The narrow bandwidth of Lora RF technology does not present any challenge for WSN applications.

In order to attend to more farming scenarios, Cloud and local dashboards are available to present data for the users. In contrast, a Cloud dashboard developed in the Grafana environment allows users to use advanced tools. The local dashboard uses the NodeRED environment, not offering advanced tools for graphics analysis and customization options. On the other hand, it offers local access to data for farms that do not have an internet connection.

ML models enable real-time monitoring the quality of collected data, searching for outliers or errors. The ML application is necessary for reliable data, allowing a more secure data-driven approach. The ML model’s low requirements allow deploying an independent Isolation Forest classifier model for each node. The low-computation requirement of the ML application allows the scalability of sensors nodes and the IoT platform. In addition, CV applications could real-time detect diseases in strawberry farming on edge. In addition, this approach allows more farmers to use Vision applications in the farming process. The model can be adjusted through a transfer learning approach, allowing users to quickly modify and deploy other detection models. The model performance fulfills the requirement for disease detection in stable objects.

The 3D-printed case protects the sensor and collector node hardware. In addition, PLA material mitigates the environmental impact of IoT platform implementation. Green renewable sources such as starch, corn, or sugar cane create PLA filament. This means that it is better for the environment than other materials, as it can be recycled. For future research, a rover or drone can be developed to move the collector node and CV system autonomously through the crop. This creates more applications for the computer vision approach. In addition, a greenhouse can be a scenario for validating the WSN application and CV vision approach. To validate the case, the application can be deployed more often in the farming process, searching for improvement points. Finally, the ML application can be retrained for other temperature and humidity profiles.

## Figures and Tables

**Figure 1 sensors-22-05866-f001:**
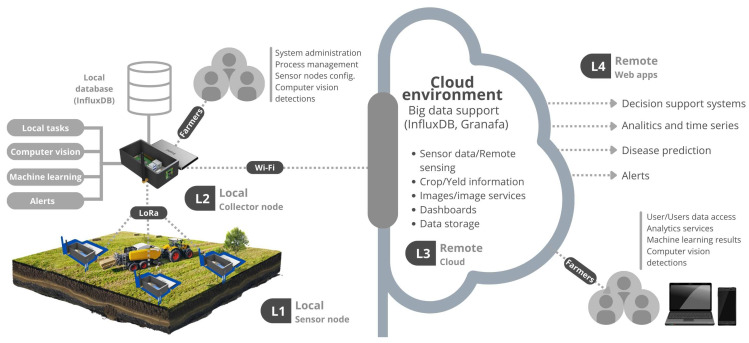
Overall IoT system architecture.

**Figure 2 sensors-22-05866-f002:**
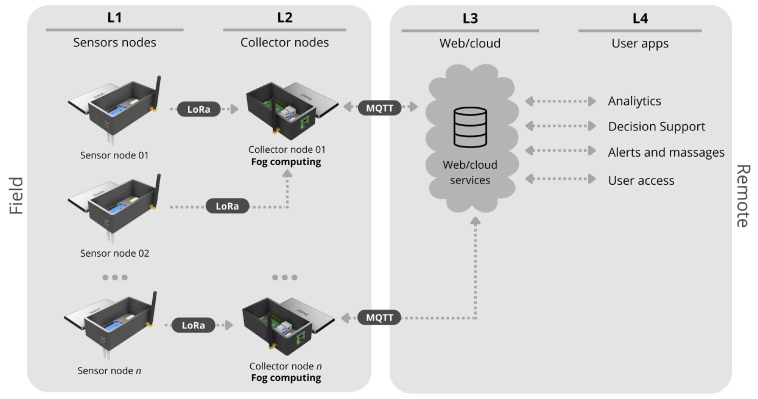
Layers of the IoT architecture.

**Figure 3 sensors-22-05866-f003:**
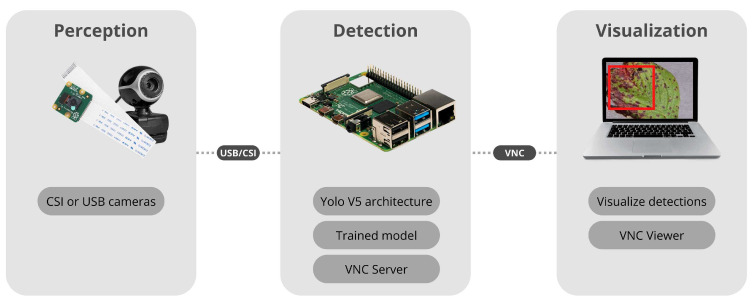
Computer vision system components.

**Figure 4 sensors-22-05866-f004:**
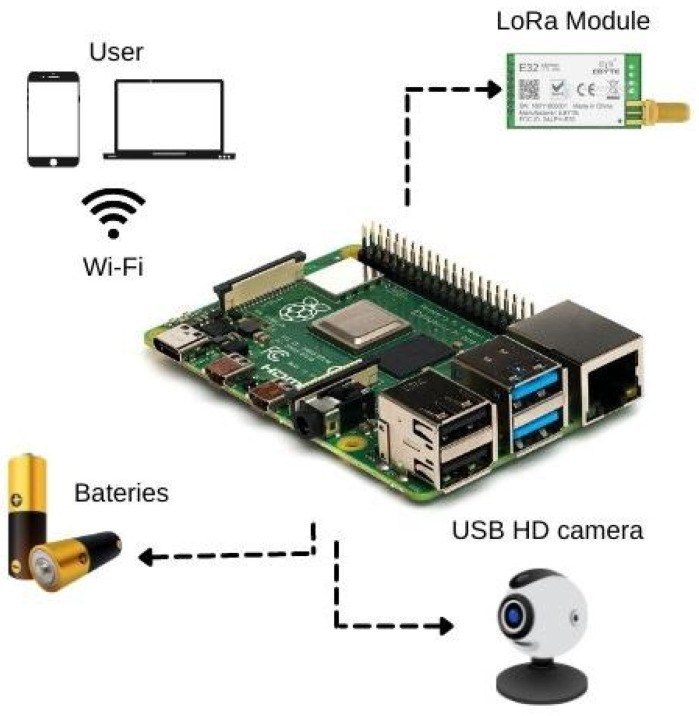
Collector node components.

**Figure 5 sensors-22-05866-f005:**
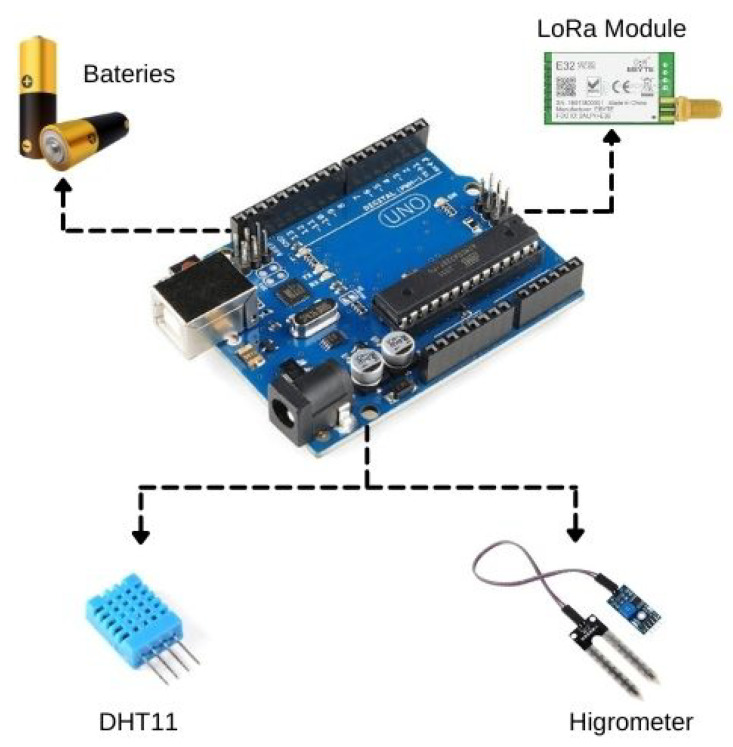
Sensor node components.

**Figure 6 sensors-22-05866-f006:**
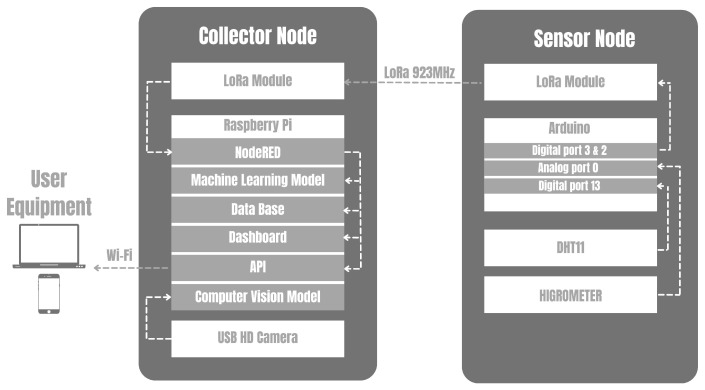
Software architecture.

**Figure 7 sensors-22-05866-f007:**
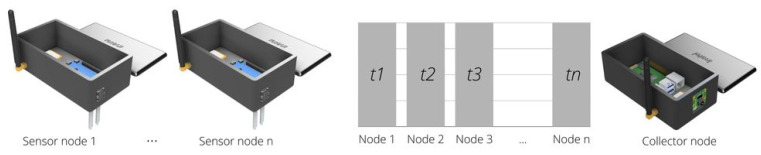
Multiplexing schema used with the IoT platform.

**Figure 8 sensors-22-05866-f008:**
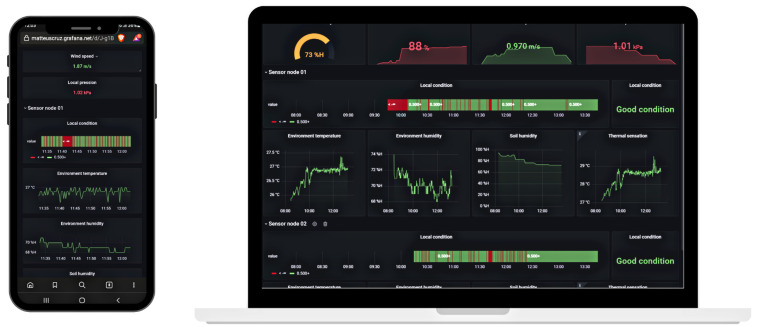
Grafana dashboard interface captured from the smartphone and laptop.

**Figure 9 sensors-22-05866-f009:**
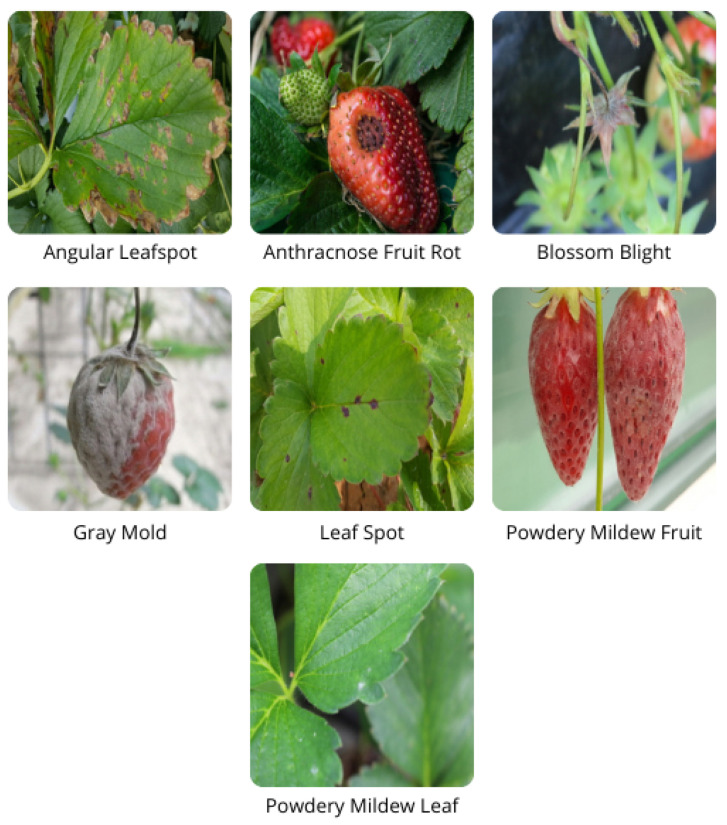
Samples of the diseases that the model is able to detect [42].

**Figure 10 sensors-22-05866-f010:**
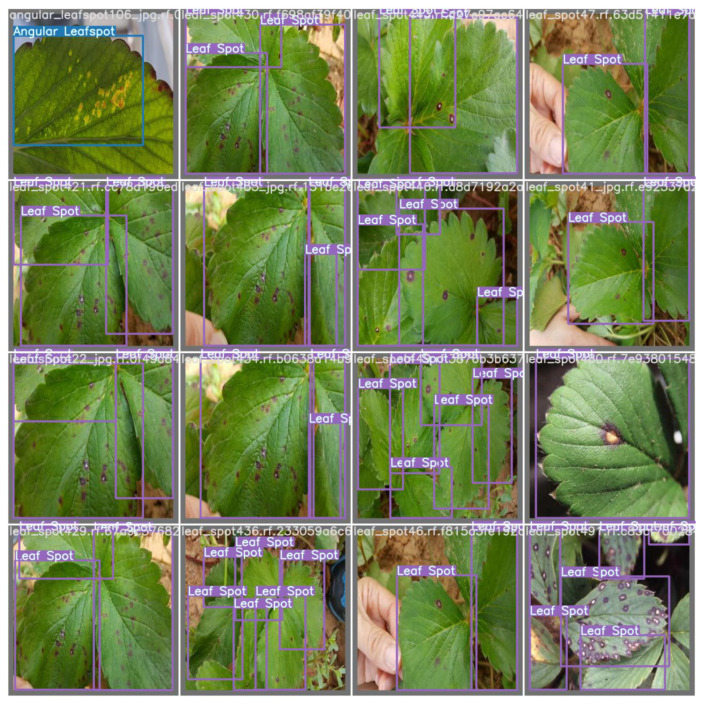
Ground truth data used to train the model.

**Figure 11 sensors-22-05866-f011:**
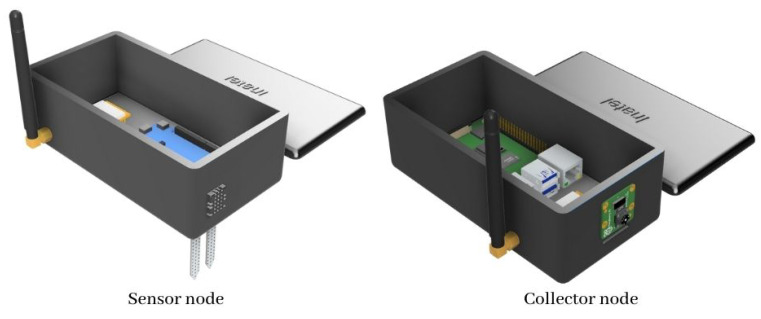
Sensor node and collector node digital design.

**Figure 12 sensors-22-05866-f012:**
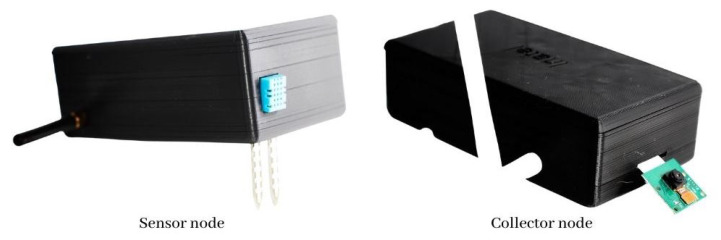
Sensor and collector nodes printed.

**Figure 13 sensors-22-05866-f013:**
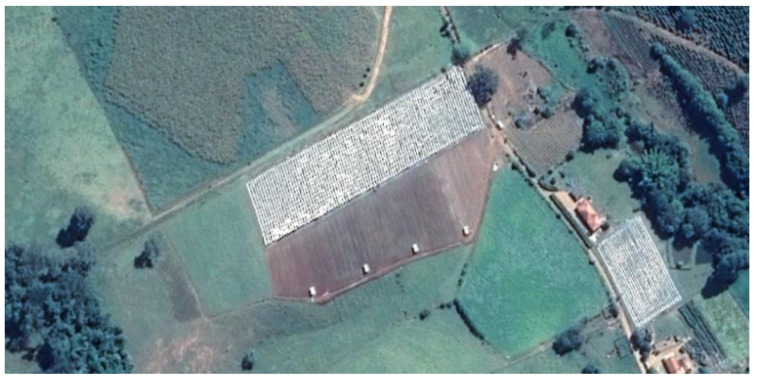
Total area of the plantation.

**Figure 14 sensors-22-05866-f014:**
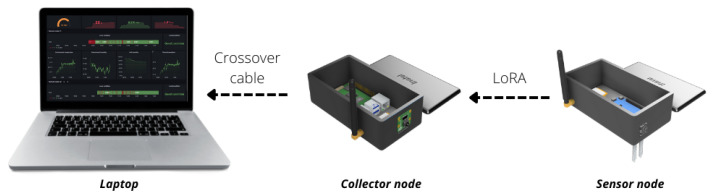
Platform connection schema used in strawberry farm.

**Figure 15 sensors-22-05866-f015:**
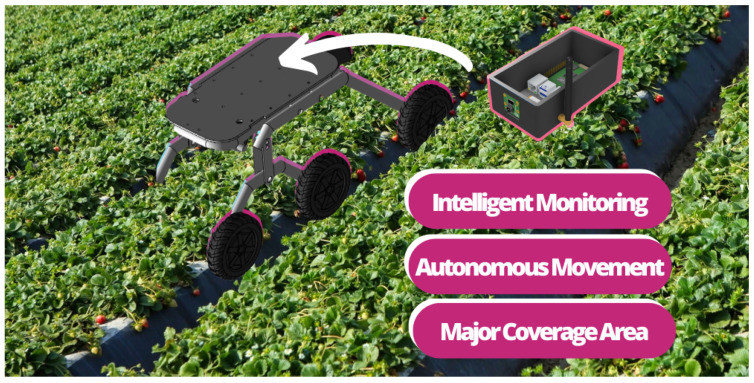
Possible application of the collector node.

**Figure 16 sensors-22-05866-f016:**
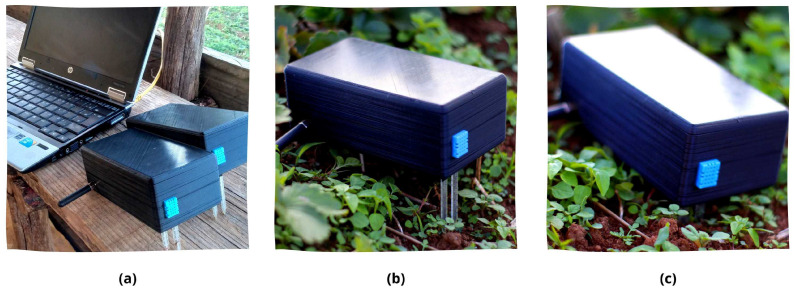
(**a**) Two sensor nodes and the computer used to present the dashboard, (**b**) sensor node positioned, and (**c**) sensor node deployed on the plantation.

**Figure 17 sensors-22-05866-f017:**
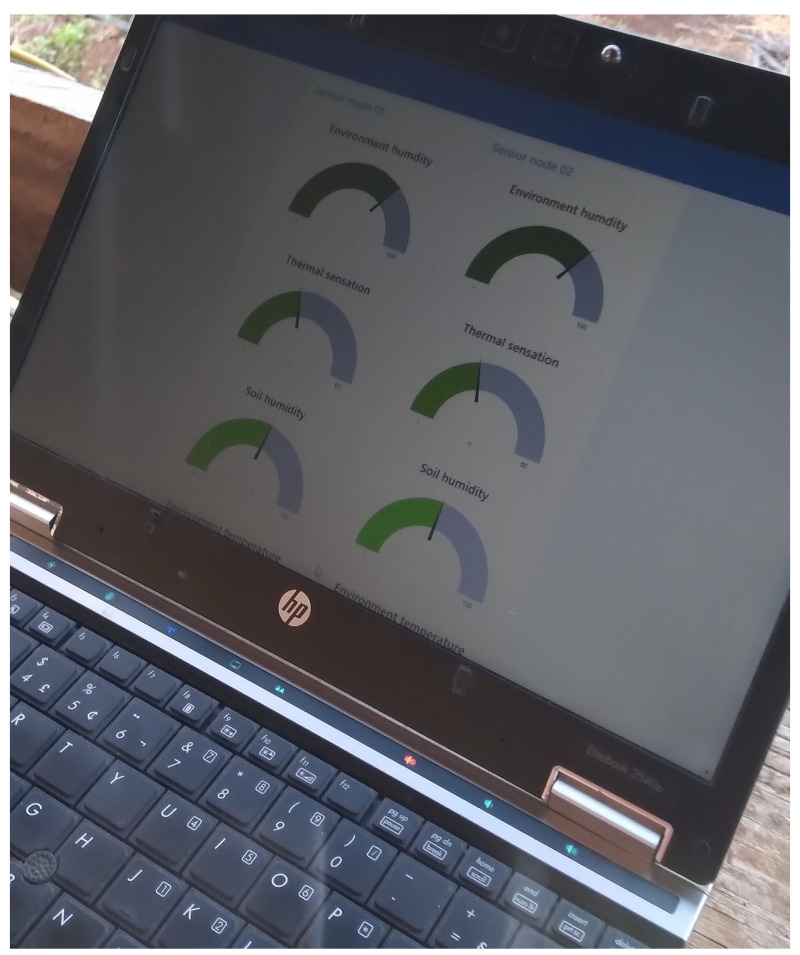
Offline dashboard applied on NodeRed.

**Figure 18 sensors-22-05866-f018:**
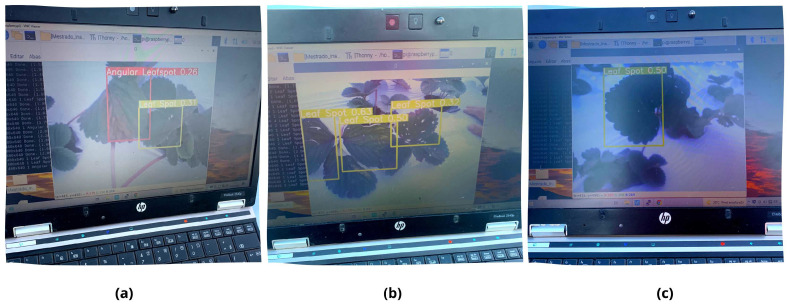
(**a**) Local detection of Angular Leaf Spot and Leaf Spot diseases, (**b**) multiple Leaf Spot diseases, and (**c**) single Leaf Spot.

**Figure 19 sensors-22-05866-f019:**
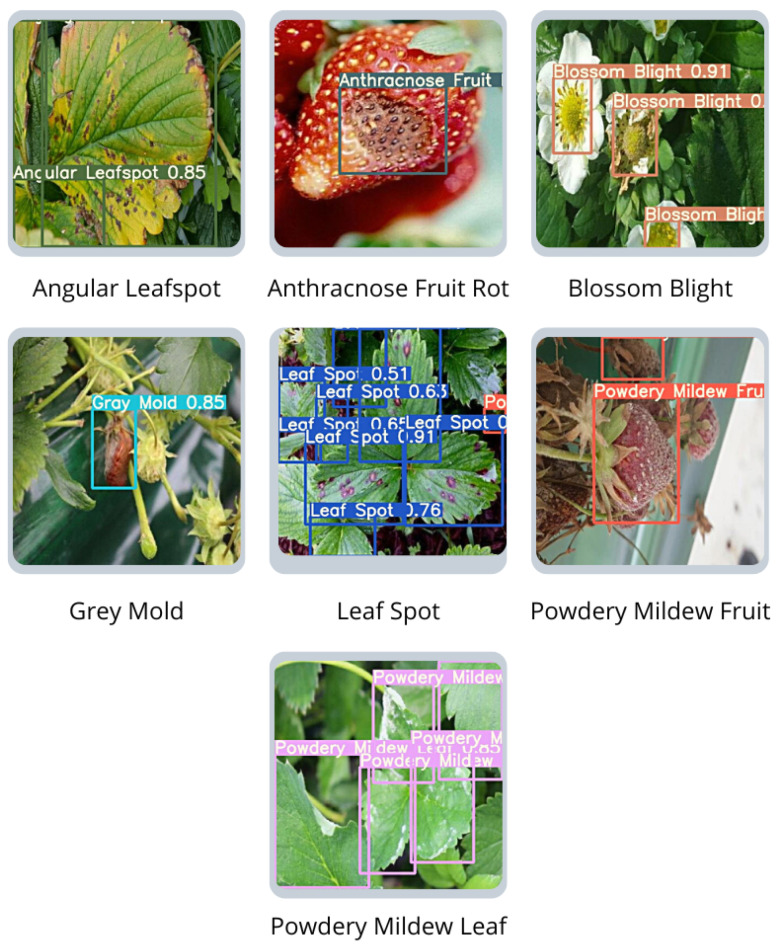
Detected diseases with Yolo v5 model.

**Figure 20 sensors-22-05866-f020:**
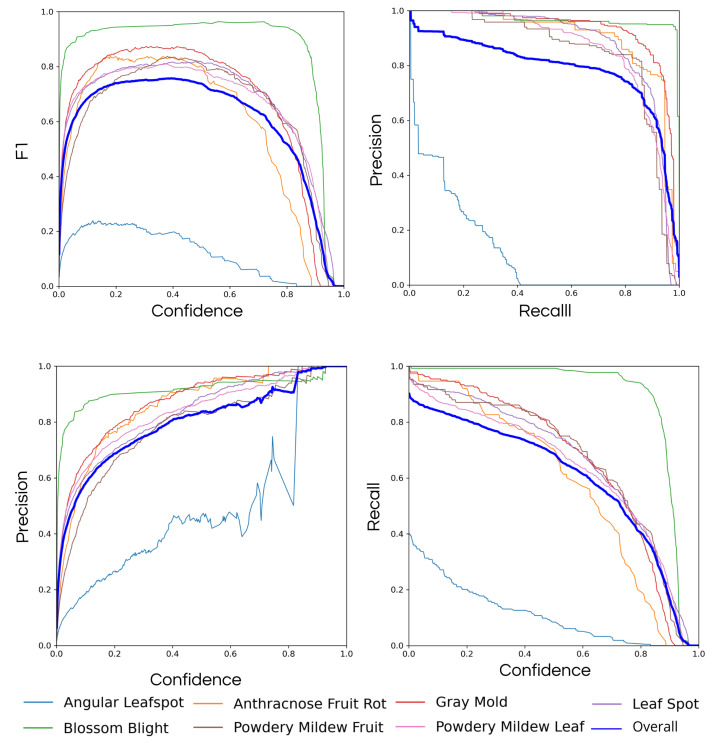
Model performance on the test set.

**Figure 21 sensors-22-05866-f021:**
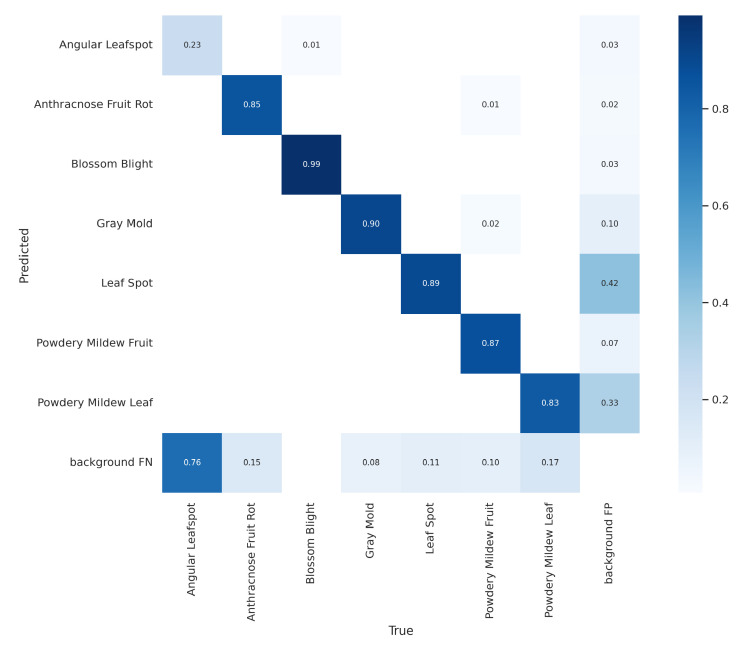
Confusion matrix obtained on the test dataset.

**Figure 22 sensors-22-05866-f022:**
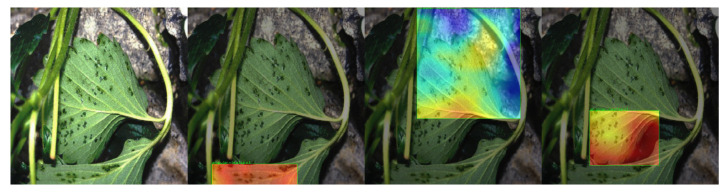
Grad-CAM test on Angular Leafspot sample.

**Figure 23 sensors-22-05866-f023:**
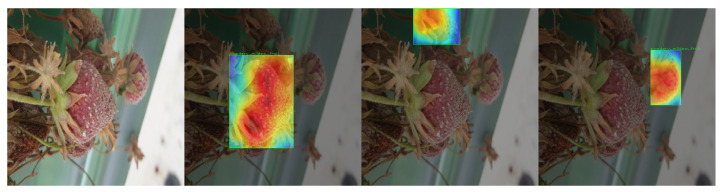
Grad-CAM test on Powdery Mildew Fruit sample.

**Figure 24 sensors-22-05866-f024:**
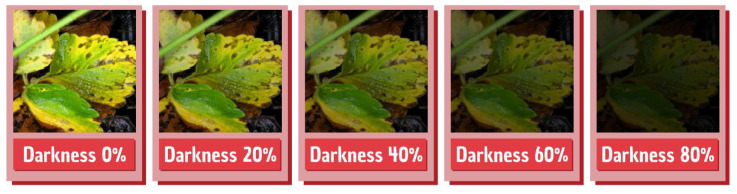
Brightness adjustments performed on the dataset.

**Figure 25 sensors-22-05866-f025:**
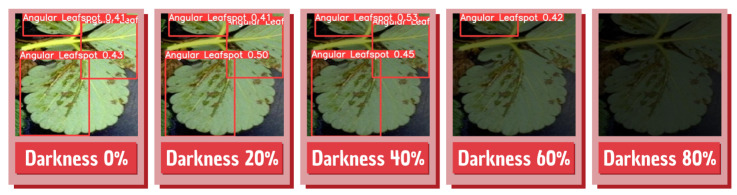
Detections performed in different brightness profiles.

**Figure 26 sensors-22-05866-f026:**
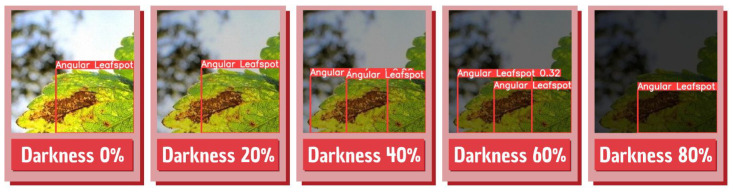
Successful detections performed in different brightness profiles.

**Figure 27 sensors-22-05866-f027:**
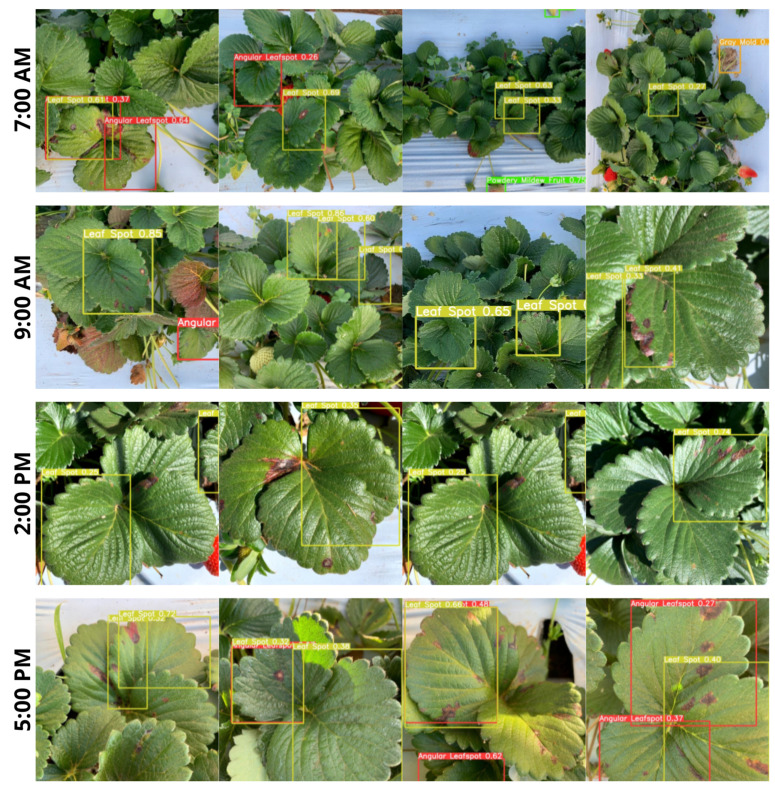
Detections performed by the model on images captured in the field.

**Figure 28 sensors-22-05866-f028:**
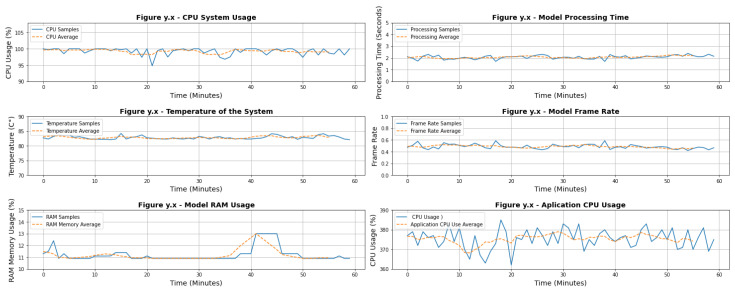
Benchmarking metrics captured during the tests performed.

**Figure 29 sensors-22-05866-f029:**
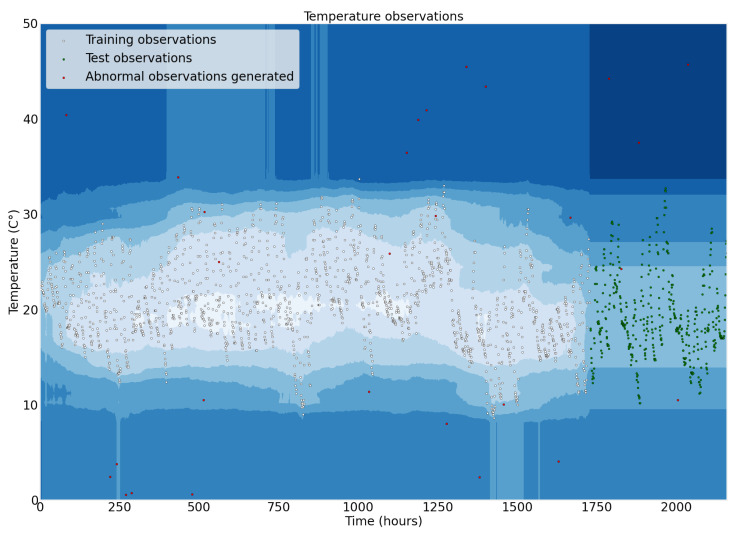
Temperature observations used to train and test the Isolation Forest model.

**Figure 30 sensors-22-05866-f030:**
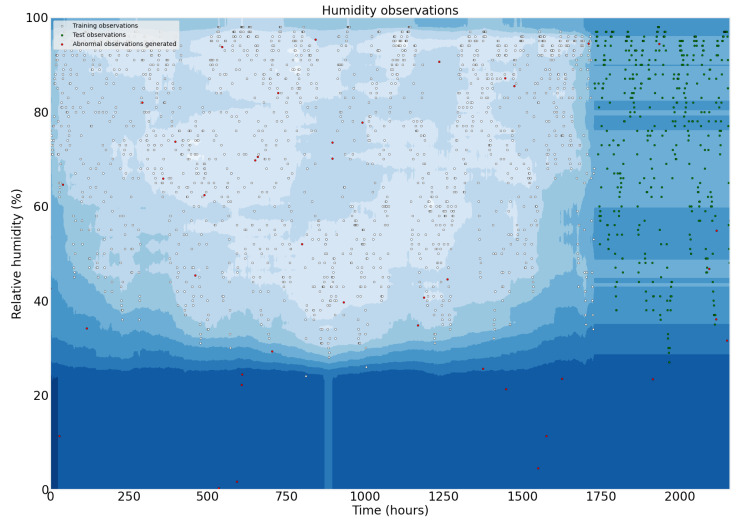
Humidity observations used to train and test the Isolation Forest model.

**Figure 31 sensors-22-05866-f031:**
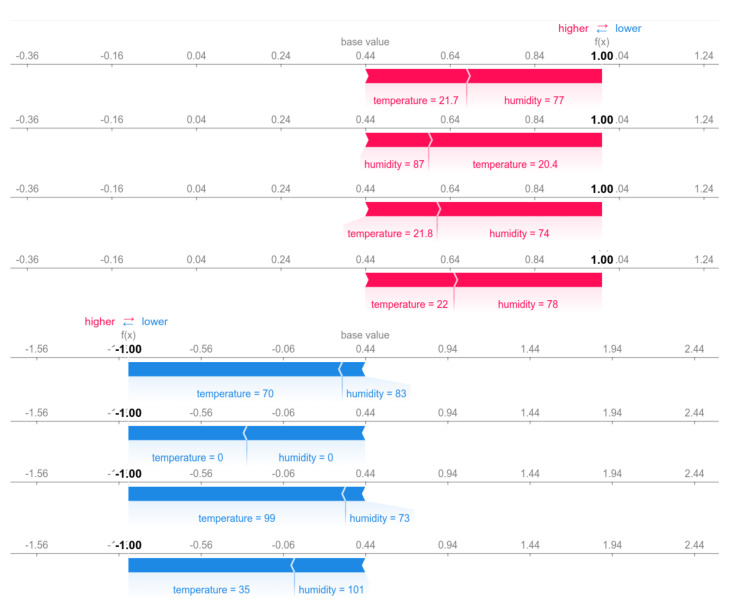
Classifications and their respective impact in the model output.

**Table 1 sensors-22-05866-t001:** Yolo v5 weights details [43].

Weight Size	Weight Name	mAP	Speed (ms)	FLOPS (MB)
Nano	YOLOv5n	28.4	6.3	4
Small	YOLOv5s	37.2	6.4	14
Medium	YOLOv5m	45.2	8.2	41
Large	YOLOv5l	48.8	10.1	89
XLarge	YOLOv5x	50.7	12.1	166

**Table 2 sensors-22-05866-t002:** List of components and items needed for the development of the platform.

Hardware	Quantity	Price
Raspberry Pi 4B	1	$107.00
Micro SD card 16 GB	1	$18.00
Pi Camera	1	$25.00
Batteries	3	$15.00
Arduino Board	2	$28.00
DHT11	2	$3.00
LoRa Module	3	$25.00
PLA Filament	1	$20.00
		$241.00

Values from Digikey.com. Access date: 25 July 2022.

**Table 3 sensors-22-05866-t003:** List of software and tools needed to run the functionalities in the platform.

Software	Price
Raspbian OS	Free
NodeRED	Free
Yolo v5	Free
Python	Free
Grafana	Free with Limitations
InfluxDB	Free with limitations
Autodesk Fusion 360	$60 month
Ultimaker Cura	Free

**Table 4 sensors-22-05866-t004:** Results obtained from the different weight sizes.

Weight Size	mAP 0.5	mAP 0.5:0.95	Precision	Accuracy	Recall	Box Loss
Yolo v5s	0.9465	0.687	0.8729	0.9074	0.9123	0.00293
Yolo v5m	0.9642	0.768	0.909	0.9021	0.9286	0.00196
Yolo v5l	0.967	0.766	0.916	0.937	0.9295	0.001896
Yolo v5x	0.975	0.771	0.920	0.954	0.9524	0.001696

**Table 5 sensors-22-05866-t005:** Training Sample Size for Different Classes.

Class	Training Sample Size
Angular Leafspot	245
Anthracnose Fruit Rot	52
Blossom Blight	117
Gray Mold	255
Leaf Spot	382
Powdery Mildew Fruit	80
Powdery Mildew Leaf	318

**Table 6 sensors-22-05866-t006:** Model performance in different brightness profiles.

Darkness Level	mAP 0.5	mAP 0.5:0.95	Precision	Accuracy	Recall
0%	0.9369	0.675	0.869	0.9112	0.951
20%	0.9324	0.762	0.919	0.9074	0.9178
40%	0.8915	0.727	0.872	0.8621	0.8562
60%	0.6256	0.623	0.697	0.6517	0.6515
80%	0.2582	0.202	0.351	0.3954	0.3174

## Data Availability

Publicly available datasets were analyzed in this study. This data can be found here: https://www.kaggle.com/datasets/usmanafzaal/strawberry-disease-detection-dataset (accessed on 14 June 2022).

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
