# Peer review of "Smart Strawberry Farming Using Edge Computing and IoT"

_sensors, 2022, doi:10.3390/s22155866_

Round 1

Reviewer 1 Report

The paper "An LoRa IoT Platform for Smart Strawberry Farming with Yolov5 and Isolation Forest" is nice and an original example of modern, intelligent sensors which have been used in conjunction with machine leaning.  The paper is of good quality and of interest to the reader.  A proof read by a native English writer would enhance the overall read. As an open journal it would seem to be important to provide source code from a GitHub (or similar).

Finally this paper is in my opinion suitable for publication with little change.

Author Response

First of all, we are grateful to the Editor and the anonymous Reviewers for their valuable time and for providing many constructive comments and suggestions which contributed to considerably improving our manuscript.

After being carefully and thoroughly revised, the manuscript has undergone the following modifications that the Reviewers recommended:

We modified the organisation of the manuscript in order to emphasize the contributions of the proposed scheme and make the manuscript more concise;
We have carefully revised it to improve the reader’s experience throughout the article and also corrected several typos;
We created a repository on GitHub and uploaded the nodes used for processing, storing, and visualizing the collected data, and added the YoloV5 network and also the weights of the model trained for disease detection in strawberry.

In the revised manuscript we highlighted in blue the changes for your convenience.

Reviewer 2 Report

Please find the comments in the attached file.

Author Response

First of all, we are grateful to the Editor and the anonymous Reviewers for their valuable time and for providing many constructive comments and suggestions which contributed to considerably improve our manuscript.
After being carefully and thoroughly revised, the manuscript has undergone the following modifications that the Reviewers recommended:
• We clarified several questions and doubts of the Reviewers, which led to some changes in the manuscript:
We modified the organisation of the manuscript in order to emphasize the contributions of the proposed scheme and make the manuscript more concise;
We have included new tables and images throughout the article to improve the reader’s experience and understanding of the proposed platform;
We have inculcated a new Section to deal specifically with Related Work.   In the revised manuscript we highlighted in blue the changes for your  convenience. In addition, we have denoted the references of the response document by using a letter “R” before the reference number, i.e. “[R1]” instead of “[1]”. Moreover, the figures in the response document are denoted as “Figure Nº”, while the figures in the manuscript are denoted as “Fig. Nº”.      

Reviewer 3 Report

Overview:

This paper presents an edge IoT platform for smart strawberry farming. The system enables collection, analysis, prediction, and detection of heterogeneous data. Experimental results validate the effectiveness of the proposed system.

Strengths

+ This paper proposes a concise platform for strawberry farming, which can be used by agriculture practitioners.

+ The case study shows that the proposed platform is deployable.

Weaknesses

- The evaluation of the CV system is not entirely convincing.

* First, the authors seem only report the performance of YOLOv5 on the training set (Line 580). This does not follow the standard procedure of computer vision, where results on the test dataset should also be reported.

* Second, Figure 19 only shows the detection results on the strawberry dataset. The reviewer would like to see more results on real scenes, i.e., images captured in the plantation area.

* Finally, I am curious about the performance of the CV model under different illumination conditions, because real scenes often suffer from sunlight changes. This reminds me of the recent advance in wheat head detection [1, 2] that tackles illumination changes. The CV model could be further improved by deploying recent progress. The authors may discuss them.

- No analysis of the time consumption of each component. Since the paper develops an edge platform, it could be insightful if the authors can delineate the time consumption of each part.

- The writing could be improved.

* missing reference on flow information (Page 4, Line 144)

* the usage of “LoRa” and “Lora” could be unified

* Typos, e.g., wight (Page 20, Table), duplicate “under” (Page 20, Line 585)

References:

[1] David, Etienne, et al. "Global wheat head detection 2021: an improved dataset for benchmarking wheat head detection methods." Plant Phenomics 2021.

[2] Liu, Chengxin, et al. "Dynamic Color Transform Networks for Wheat Head Detection." Plant Phenomics 2022.

Author Response

First of all, we are grateful to the Editor and the anonymous Reviewers for their valuable time and for providing many constructive comments and suggestions which contributed to considerably improve our manuscript.
After being carefully and thoroughly revised, the manuscript has undergone the following modifications that the Reviewers recommended:
• We clarified several questions and doubts of the Reviewers, which led to some changes in the manuscript:
We modified the organization of the manuscript to emphasize the contributions of the proposed scheme and make the manuscript more concise;
We have included new tables and images throughout the article to improve the reader’s experience and understanding of the proposed platform.   In the revised manuscript, we highlighted in blue the changes for your convenience.

Round 2

Reviewer 2 Report

I thank the authors for revising the manuscript. The contents of the article have been improved, however, there are many comments which are not addressed by authors. It is advised to authors that carefully consider each of the provided comment and justify it on its merit. Also, I have found conflict in authors response and original article, which is not a moral practice (its mentioned in the response that change has been made, however no significant modifications spotted in the revised manuscript) i.e., comment about language editing, comment about existence of short paragraphs, comment about adding more details into the dataset section, comment about adding detailed cost analysis section, comment about the merging of conclusion paragraphs. Please see the following detailed description of unaddressed comments from the first review.

1.     Comment 6, Authors were advised to revise the manuscript and remove the short paragraphs by merging them together where applicable. A standard paragraph length is 10-12 lines. No considerable action taken in this regard. 

2.     Comment 9, Authors were advised to combine the paragraphs to a maximum of 2 or three. It is said in the response that authors have considered the comment, however, upon comparing, I was able to find that there has been no whatsoever change made in terms of paragraphs merging. This is not a recommended academic practice and may cost you penalty in future. Either you address a comment to its merit, or you provide a rebuttal why you can’t do that. By just saying it done, it is often considered as misleading. 

3.     Authors were advised to get the manuscript proofread by native language editor. However, only few spelling and minor changes can be spotted in the revised version. Language editing may result in major sentence, vocabulary and transition level changes. Not satisfied with the effort made in this regard. 

4.     Comment 15, author needs to answer all the asked question into the revised manuscript, specifically within the dataset section and experimental protocols section. 

5.     Comment 16: How did authors calculated the frame_rate? 

6.     Comment 17: Authors were advised to provide a detailed cost analysis of the proposed system. The response of authors is nowhere incorporated in the revised manuscript, authors need to make sure that responses are also reflected in the revised version. Also, it seems like only the hardware cost has been listed, by detailed cost analysis reviewer meant all the costs involved in the development of such solution, instalment and operational maintenance. This may include software, logistic, cloud, app development and other costs as well. 

Author Response

First of all, we are grateful to the Editor and the anonymous reviewers for their valuable time and for providing many constructive comments and suggestions which contributed to improve considerably our manuscript.
After being carefully and thoroughly revised, the manuscript has undergone the following modifications that the Reviewers recommended:

• We clarified several questions and doubts of the Reviewers, which led to some changes in the
manuscript:

– We modified the organization of the manuscript to emphasize the contributions of the proposed scheme and make the manuscript more concise;

In the revised manuscript, we highlighted the changes for your convenience in blue

Reviewer 3 Report

The authors have addressed most of my concerns. However, I still have some minor concerns. Additional comments are listed as follows:

1. The test performance of different diseases is diverse. As shown in the paper, the performance of the Angular Leafspot disease is particularly low compared with other diseases. Further analysis of this detection failure could be insightful. 

2. The authors show that different illumination conditions have a significant impact on test performance. The authors shall indicate how to address this issue in the paper, such as discussing the possible solution and citing the relevant papers (e.g., papers mentioned in my previous comments).

3. The paper still contains some typos (e.g.,  Line 667: .. -> .). The authors should check the paper thoroughly.

Author Response

First of all, we are grateful to the Editor and the anonymous reviewers for their valuable time and for providing many constructive comments and suggestions which contributed to improve considerably our manuscript. After being carefully and thoroughly revised, the manuscript has undergone the following modifications that the Reviewers recommended:

• We clarified several questions and doubts of the Reviewers, which led to some changes in the
manuscript:

– We modified the organization of the manuscript to emphasize the contributions of the proposed scheme and make the manuscript more concise;
– We have included new tables and images throughout the article to improve the reader’s experience and understanding of the proposed platform.

In the revised manuscript we highlighted in blue the changes for your convenience.
